# Biochemical Characterisation of the Short Isoform of Histone N-Terminal Acetyltransferase NAA40

**DOI:** 10.3390/biom14091100

**Published:** 2024-09-02

**Authors:** Ariel Klavaris, Maria Kouma, Cem Ozdemir, Vicky Nicolaidou, Kyle M. Miller, Costas Koufaris, Antonis Kirmizis

**Affiliations:** 1Epigenetics and Gene Regulation Laboratory, Department of Biological Sciences, University of Cyprus, Nicosia 2109, Cyprus; klavaris.ariel@ucy.ac.cy (A.K.); kouma.maria@ucy.ac.cy (M.K.); 2Department of Molecular Biosciences, The University of Texas at Austin, Austin, TX 78712, USA; cem.ozdemir@utexas.edu (C.O.); kyle.miller@austin.utexas.edu (K.M.M.); 3Department of Life Sciences, University of Nicosia, Nicosia 2417, Cyprus; nicolaidou.v@unic.ac.cy; 4Cyprus Cancer Research Institute, Nicosia 2109, Cyprus

**Keywords:** NAA40, isoform, testis, enzymatic assay

## Abstract

N-alpha-acetyltransferase 40 (NAA40) is an evolutionarily conserved N-terminal acetyltransferase (NAT) linked to oncogenesis and chemoresistance. A recent study reported the generation of a second, shorter NAA40 isoform (NAA40^S^) through alternative translation, which we proceeded to further characterise. Notably, recombinant NAA40^S^ had a greater in vitro enzymatic activity and affinity towards its histone H2A/H4 substrates compared to full-length NAA40 (NAA40^L^). Within cells, NAA40^S^ was enzymatically active, based on its ability to suppress the H2A/H4S1Ph antagonistic mark in CRISPR-generated *NAA40* knockout cells. Finally, we show that in addition to alternative translation, the NAA40^S^ isoform could be derived from a primate and testis-specific transcript, which may align with the “out-of-testis” origin of recently evolved genes and isoforms. To summarise, our data reveal an even greater functional divergence between the two NAA40 isoforms than had been previously recognised.

## 1. Introduction

N-terminal acetylation (Nt-Ac) is an abundant eukaryotic protein modification deposited by evolutionarily conserved N-terminal acetyltransferase (NAT) complexes [1]. One of these is the N-alpha-acetyltransferase 40 (NAA40 or NatD), a highly specific NAT that Nt-acetylates histones H2A/H4 [2]. Importantly, Nt-Ac of H2A/H4 antagonises the deposition of other histone modifications—specifically H2A/H4S1Ph and H4R3me—to influence gene regulation [3,4]. Consistent with non-redundant and biologically important functions for NAA40, it has been implicated in oncogenesis and chemoresistance [5,6,7]. There is thus an increasing interest in investigating the regulatory and molecular mechanisms through which this NAT exerts its functions.

Beyond the deep evolutionary conservation of eukaryotic NAT, alternative splicing, variable promoter usage, differential transcript termination, and alternative translation initiation are additional mechanisms that expand the proteomic diversity of eukaryotic genes. For NAA40, it was recently reported that alternative translation initiation generates a shorter NAA40 isoform (NAA40^S^) [8]. In their comparison of the NAA40^S^/NAA40^L^ isoforms, the authors stated that the shorter isoform was primarily cytosolic, while the longer was primarily nuclear. Consequently, alternative translation initiation is one mechanism for increasing NAA40 proteomic diversity. Although alternatively spliced transcripts have been reported for other NATs [9,10,11], the potential contribution of alternative splicing to the regulation and generation of NAA40 protein isoforms has not been previously examined. Here, we focused on further characterising the NAA40^S^ isoform. The application of biochemical and cellular assays revealed novel insights about functional differences between these two NAA40 isoforms. Additionally, we report here that beyond alternative translation, the NAA40^S^ isoform is also encoded by a testis-specific transcript.

## 2. Materials and Methods

### 2.1. Data Acquisition and Processing

*NAA40* transcript sequences were extracted from Ensembl Release 104 (May 2022). Scanprosite (https://prosite.expasy.org/scanprosite/) (accessed on 1 June 2022) was used to screen for the GNAT motif PS51186 and the Q/RxxGxG/A acetyl-CoA motif. Processed *NAA40* transcript levels (TOIL RSEM isoform percentage) for the Genotype-Tissue Expression (GTEx) and Cancer Genome Atlas (TCGA) were obtained from the UCSC Xena database (https://xenabrowser.net/datapages/) (accessed on 1 July 2022). Transcripts Per Million (TPM) for the Human Protein Atlas were obtained from the study’s portal (https://www.proteinatlas.org/about/download) (accessed on 1 July 2022), followed by a calculation of isoform percentages through the summation of transcript TPM values.

### 2.2. Polymerase Chain Reaction (PCR)

RNA from non-pathological human liver (Catalogue No: R1234149-50, 64-year-old male) and testis (Catalogue No: R1234260-50, 27-year-old male) samples were obtained from Biochain. HCT116 RNA was isolated using the RNeasy Mini kit (Qiagen, Hilden, Germany) and cDNA generated using the High-capacity cDNA reverse transcription kit (Applied Biosystems, Waltham, MA, USA). PCR was performed using the KAPA Taq PCR kit (KAPA Biosystems, Wilmington, MA, USA) and the products were separated in 2% agarose gels. Primers were ordered from IDT: *NAA40*^S^ F-AAATATGATAGAAACGGCACACAGA, R-AGCAATCCTCCCCACAGCAA; *NAA40*_124_ F-AATATGATAGAAACGGCACACAGA, R-CCAGTCCAGACACTCGCTTA; *NAA40*FL F-GAAGAAGCTGAGCGTTGTCG, R-ACTTCTCCCCAGGGTTCTGT; *NAA40*tot F-CCTTCGACCTGACCAAAACG, R-ACGTCAAACCGGAAGTGAGA.

### 2.3. Protein Expression and Purification

The *NAA40*^S^ cDNA plasmid was obtained from Origene, Rockville, MD, USA (CAT#: RC236637) and the *NAA40*^L^ plasmid was described previously [12]. For recombinant protein expression/purification cDNA encoding, the two isoforms were cloned into pGEX-5x-1 and validated using DNA sequencing (Macrogen, Amsterdam, The Netherlands). The pGEX-5x-1 plasmids were used to transform BL21 (DE3) codon plus RIL pACYC *E. coli* cells, which were grown for 16 h at 37 °C in 40 mL of LB medium, supplemented with 100 μg/mL of Ampicillin (Sigma A9518, St. Louis, MO, USA) and 25 μg/mL of Chloramphenicol (Sigma C0378-5G, St. Louis, MO, USA). Final culture volume was adjusted to 400 mL, and after 2 h (OD 600 nm = 0.4), protein expression was initiated with 0.1 mM of isopropyl β-D-1-thiogalactopyranoside (IPTG) (Thermo Fischer 15529019, Waltham, MA, USA). After a further 4 h, pelleted bacteria were resuspended in ice cold PBS/1% Triton X-100 supplemented with protease inhibitors. Cells were lysed by small-probe sonication (Amplitude 30%, 3 × 45 s on ice), and the supernatant was incubated with Glutathione agarose beads (Cytiva 17075601, Marlborough, MA, USA) and washed with ice-cold PBS/1% Triton X-100. Beads with the recombinant proteins were stored at −20 °C in 80% PBS/20% glycerol. To obtain the GST tagged proteins, the buffer of 50 mM of Tris HCl pH 8, 20 mM of reduced Glutathione-20% glycerol was used for elution. Factor Xa (NEB P8010S, Ipswich, MA, USA) was used to remove the GST tag from the beads and dialyzed in 50 mM of Tris-HCl pH 7.5, 100 mM of NaCl, 25% Glycerol. The identities of the recombinant proteins were confirmed using SDS-PAGE, and concentrations were determined using the BIO-RAD protein assay dye reagent (BIO-RAD 5000006, Hercules, CA, USA and a Nanodrop ND-1000.

### 2.4. Fluorescence-Based In Vitro Enzymatic Assay

We adapted a reported NAA40 fluorescence assay [13] to assess isoform kinetics against synthetic histone H4 (1–8) peptide (BIOSYNTAN GmbH, Berlin, Germany, >95% purity). Peptide concentration varied from 0.1 μM to 25 μM (1:2 dilutions). The reactions were carried out in triplicates, and the assay was repeated five times (n = 5) and contained 15 μM of Thioglo4 (Merck 595504, Darmstadt, Germany), 10 μM of Acetyl coenzyme A lithium salt (SANTA CRUZ sc-214465, Dallas, TX, USA), 25 mM of HEPES pH 7.5, 150 mM of NaCl, 0.01% Triton X-100) and 50 nM of NAA40 (long or short) in 40 μL reactions. The addition of substrates-initiated reactions and fluorescence was monitored with a Tecan Spark multimode reader with excitation 410 ± 20 nm and emission 475 ± 20 nm for 5 min. The standard curve y = 915.6 × X − 52.90 was generated using 0–10 μM of 1:2 dilutions of Coenzyme A sodium salt hydrate (Merck C13144, Darmstadt, Germany) and 25 mM of HEPES pH 7.5, 150 mM of NaCl, and 0.01% Triton X-100.

### 2.5. Generation of NAA40^−/−^ Cell Line Using CRISPR

A set of guide RNA (gRNA) designed to cleave a genomic region within exons 4 and 5 of human *NAA40* (F1′ CACCGTCCGGTTTGACGTGGAGTGT; R1′ AAACACACTCCACGTCAAACCGGAC; F2′ CACCGTCGAAGGCCCAATCCACGG; R2′ AAACCCGTGGATTGGGCCTTCGAC) and No Template Control (NTC) gRNA (F′ CACCGGTATTACTGATATTGGTGGG; R′ AAACGTATTACTGATATTGGTGGGC) were cloned into pSpCas9 vector (Addgene #62988, Watertown, MA, USA) and were used to transfect HCT116 cells using Fugene HD (Promega #E2311, Madison, WI, USA) following the manufacturer’s instructions. Transfected cells were Puromycin selected and *NAA40* screen primer pairs (F′ TGTTCCTGCACCTTCTCTAGTC; R′ GAGCTTGCAGTCCTTGGCA) were used to amplify ~600 bp region in WT alleles and ~350 bp in knockout (KO) alleles in the *NAA40* gene. Colonies showing a single PCR band at ~350 bp detected using ethidium staining in agarose gel electrophoresis were validated by the increase in H2A/H4S1Ph (Abcam #ab222765, Cambridge, UK) using Western blotting. Cell colonies derived from cells transfected with non-targeting gRNA exhibited a single DNA band ~600 base pairs in size and showed no increase in H2A/H4S1ph. Subsequent experiments were conducted on single-validated KO or control cell lines.

### 2.6. Mammalian Cell Culture Experiments

HCT116 colorectal cancer cells were maintained as previously described [12]. Transient transfections of 1 µg of cDNA plasmids was performed using a Lipofectamine 3000 (Thermo Fischer, Waltham, MA, USA).

### 2.7. Protein Extraction

Whole cell protein extracts were collected using a Lysis Buffer (50 mΜ of Tris-HCl pH 8, 3 mM of EDTA, 100 mM of NaCl, 1% Triton X-100, 10% glycerol, 0.5 mM of PMSF, protease inhibitor cocktail). For biochemical fractionation, 1 × 10^6^ cells harvested in 1× PBS were lysed in Buffer S (10 mM of HEPES, 10 mM of KCl, 1.5 mM of MgCl_2_, 0.34 mM of sucrose, 10% glycerol) plus 0.1% Triton X-100 and 1× protease inhibitor cocktail) on ice followed by centrifugation to separate the fractions. To purify the nuclear fractions, pellets were re-suspended in a low-salt lysis buffer (20 mM of Hepes, 1.5 mM of MgCl_2_, 150 mM of NaCl, 0.2 mM of EDTA, 25% glycerol) plus 0.1% *v/v* NP-40, 0.2 mM of DTT and protease inhibitors (0.5 mM of PMSF and 1× protease inhibitor cocktail) and treated with 200 units/mL of benzonase (Sigma E1014) at 37 °C.

### 2.8. Immunoblotting

For immunoblotting, 20 μg of protein extract was separated using SDS-PAGE, transferred to a nitrocellulose membrane (GE Healthcare, Chicago, IL, USA), blocked with 3% TBS-T/BSA for 1 h at RT, and incubated overnight with primary antibodies at 4 °C. The antibodies used were H2A (1:1000; 07-146, Sigma-Aldrich, St. Louis, MO, USA), ACTIN (1:1000; ab8227, Abcam, Cambridge, UK), H4/H2AS1ph (1:1000; ab177309, Abcam, Cambridge, UK), GAPDH (1:2500; ab9485, Abcam, Cambridge, UK), PATT1 (1:1000; ab106408, Abcam, Cambridge, UK), H4 (1:1000; 05-858 Millipore, Darmstadt, Germany), H3 (1:1000; ab1791, Abcam, Cambridge, UK), Glutathione S-transferase (GST) (1:10,000; A7340, Sigma, St. Louis, MO, USA), and PCNA (1:1000; ab29, Abcam, Cambridge, UK). For the secondary antibody, a horseradish peroxidase (HRP)-conjugated goat anti-rabbit IgG (1:30,000, Thermo Fischer, Waltham, MA, USA) and a polyclonal goat anti-mouse immunoglobulins/HRP (1:1000, Invitrogen, Waltham, MA, USA) were used. Bands were detected using the enhanced chemiluminescence ChemiDoc system (BIO-RAD, Hercules, CA, USA).

### 2.9. GST-Pull Down

The GST-pull down assay was as outlined by [14]. HCT116 nuclear lysates were incubated with 25 μg of GST-tagged protein for pre-cleaning at 4 °C, with end-over mixing for 2 h. Subsequently, the lysates were incubated for an additional two hours with either GST tag (10 μg), GST-NAA40^S^, or GST-NAA40^L^ at equimolar concentrations. Four washes were carried out using a GST Lysis buffer (20 mM of Tris-HCl pH 8, 200 mM of NaCl, 1 mM of EDTA pH 8, 0.5% NP40, protease inhibitor cocktail, 25 μg/μL of PMSF) followed by elution with 20 mM of reduced glutathione in 50 mM of Tris-HCl pH 8.

## 3. Results

### 3.1. NAA40^S^ Is Encoded by a Primate-Evolved Transcript Variant

Protein isoforms are biologically interesting due to their exhibiting functional diversity and distinct regulation. We initially verified the finding [8] that two isoforms (NAA40^S^/NAA40^L^) are generated by the alternative translation of the exogenously transfected canonical transcript (Figure 1A). We next scrutinised the NCBI and Ensemble databases for human transcripts that encode potential NAA40 isoforms. This revealed two human transcripts that specifically encode NAA40^S^ but not NAA40^L^, NM_001300800 (NCBI database), and ENST00000542163 (Ensemble database). Both of these transcripts differed in their transcription start sites and 3′UTR lengths, but use an alternative first exon, 1b, and initiation of translation from exon 2 to encode the N-terminally truncated NAA40^S^ (Figure 1B). Transfection of a plasmid containing NM_001300800 into *NAA40* KO HCT116 cells proved the generation of the NAA40^S^ from transcripts utilising the alternative exon 1 (Figure 1C). Interestingly, examination of the sequence of the human exon 1b, which is included in the NAA40^S^ isoform, using the BLAST tool found highly conserved homologs in multiple primate species (e.g., chimpanzee, gorilla, macaque), while no hits were found in non-primate species. Likewise, in Ensemble transcripts encoding NAA40^S^, it could be found for primate species (Chimpanzee ENSPTRT00000104645; Gibbon ENSNLET00000048285; Macaque ENSMNET00000042303), but not in non-primate species. Thus, exon 1b is a recently evolved genomic feature.

Differential splicing or the use of different transcription start/end could generate further NAA40 protein isoforms beyond NAA40^S^/NAA40^L^. The Ensembl database lists one additional protein-coding transcript beyond those encoding NAA40^S^/NAA40^L^, ENST00000539656. This transcript is generated through an alternative splicing event that skips exons 4–6 and encodes a putative 124 amino acid isoform (Uniprot ID: F5H2C9). We validated the presence of the exon 4–6 skipping event in HCT116 cells using RT-PCR (data available upon request). However, the catalytic GNAT domain and the essential acetyl-CoA-binding motif are both missing from NAA40_124_, suggesting that it is catalytically inactive. No other potential protein coding *NAA40* transcripts were found in NCBI, thus supporting NAA40^S^/NAA40^L^ as the main enzymatically active isoforms of this NAT.

### 3.2. Biochemical Characterisation of NAA40^S^

Both NAA40^S/L^ were shown to be capable to catalyse the Nt-Ac of histone H2A in yeast [8]. However, this qualitative analysis did not allow for a direct quantitative comparison of their enzymatic activity. For this purpose, we set up an in vitro acetylation assay, as previously described [13]. Surprisingly, despite the missing N-terminal end, the NAA40^S^ isoform had a significantly greater catalytic efficiency (Kcat/Km) towards histone H4 compared to full-length NAA40 (Figure 2A). We next compared the affinity of the two isoforms towards histones using affinity purifications of GST-tagged recombinant proteins with nuclear lysates. Consistent with the enzymatic activity, these experiments revealed a clearly greater binding of GST-NAA40^S^ 25kDa compared to the longer isoform towards (Figure 2B) their known substrates H2A/H4 and with histone H3. The detection of H3 affinity is due to the interaction of the isoforms with nucleosomal H4/H2A. Consequently, in vitro characterisation revealed clear differences in the essential biochemical properties of the two NAA40 isoforms.

We next proceeded to characterise the NAA40^S^ function and localisation in human cells. For this purpose, we used CRISPR to generate *NAA40*^L/S^ knockout (KO) HCT116 cells. The gRNA that we designed flanked exons 4/5 of *NAA40*, which contain the GNAT catalytic domain. Thus, this genomic editing ensures that all potential NAA40 isoforms are rendered catalytically inactive. We validated, using PCR, the deletion of the targeted region in *NAA40* KO cells (Figure 3A). In contrast to a previous report, we observed the expression of ectopically expressed NAA40^S^ in both the nuclear and cytosolic compartments (Figure 3B). As expected according to previous findings, the antagonistic marks H2A/H4S1Ph were induced in the KO cells. Complementation of *NAA40* KO cells with NAA40^S^ was able to repress H2A/H4S1Ph, demonstrating that it is enzymatically active when produced within cells (Figure 3C).

### 3.3. Testis-and-Primate-Specific Transcript Variant Encoding NAA40^S^

The existence of transcript(s)-encoding NAA40^S^ opens the possibility of NAA40^S^ being transcriptionally regulated independently from NAA40^L^. To assess this possibility, we examined human RNA-Seq datasets from a collection of non-pathogenic—the Genotype-Tissue Expression (GTEx) and the human protein atlas (HPA)—as well cancerous tissue types found in The Cancer Genome Atlas (TCGA). Notably, this analysis highlights a highly consistent transcript landscape for the human *NAA40* gene across diverse tissue types, which is also largely unaffected in cancer cells. Consistently across examined tissues and in all the three studies, the most abundant transcripts were ENST00000377793 encoding NAA40^L^/NAA40^S^ and the non-coding ENST00000338447 that retains the intron between exons 2/3 (Figure 4A). The NAA40^S^/NAA40_124_-encoding transcripts were below 1% isoform percentage in all examined tissues, apart from the testis where ENST00000542163 (encoding NAA40^S^) is enriched to ~20% (Figure 4B). Consistent with our analysis, the amplification of transcripts containing exon 1b, which is uniquely found in NAA40^S^, was successful in a human testis sample, but not in a human liver sample (Figure 4C). We also note that the ages of the individuals from which the tissues were isolated differed (64-year-old individual contributing the liver sample vs. 27-year-old individual contributing the testis sample), although we do not consider it likely that this age difference affects this result. Thus, the involvement of transcript-mediated synthesis of NAA40^S^ appears to be restricted to the testis.

## 4. Discussion

Consistent with previous reports [8], we have shown that NAA40^S^ is generated through either alternative translation initiation or from translation of a transcript variant (ENST00000542163; NM_001300800). Notably, our analysis supports that the transcript-encoding NAA40^S^ is testis- and primate-specific. Consequently, NAA40^S^ is likely to be primarily generated by alternative translation initiation in non-testicular cells. This is the first reported instance of a NAT protein-coding transcript with a highly tissue-specific expression, although it is not currently clear if this restricted tissue expression is biologically relevant. Chromatin modifications, such as histone hyperacetylation, are key events in spermatogenesis, where replacement of histones with non-histone proteins occurs [15]. Consequently, epigenetic enzymes with testis-specific expression and involvement in spermatogenesis have been identified, such as the double bromodomain BET factors, BRDT [16] and SCML2 [17]. Recently, regulation of protein stability by NAA20 was reported to control male germline stem cell differentiation and reproduction in drosophila [18]. One potential explanation for the testis-specific expression of the NAA40^S^ transcript is that the shorter isoform has some non-redundant testis-specific function relating to distinct localisation or substrate interactions. An alternative explanation for this observation is that newly evolved exons, such as exon 1b of NAA40^S^, are predominantly included in testicular transcripts [19]. The so-called “out of testis” hypothesis proposes that the transcription of new genes or exons occurs originally in cells within this organ due to a permissive chromatin state and overexpression of the cellular transcription machinery. The subset of these newly transcribed genes/variants that are beneficial are then selected for stronger and/or more widespread tissue expression [20]. A second possibility, therefore, for the testis-specific NAA40^S^ coding transcript, is that evolution has not had enough time yet to either remove this variant or expand its expression range beyond testicular cells if it is biologically beneficial.

We next focused on characterising NAA40^S^ activity in colorectal cells, given the previous links between this NAT and colorectal cancer [12,21]. Generating *NAA40* KO cells, we were able to show that NAA40^S^ repressed the H2A/H4S1Ph mark and thus is able to at least partly compensate for the longer isoform. In terms of its cellular localisation, when expressed in HCT116, NAA40S has a strong cytosolic and nuclear presence. In contrast to our experimental findings on NAA40^S^ localisation, Jonckheere and Van Damme 2021 [8] reported primarily cytosolic localisation for the shorter isoform. The reason for this discrepancy is currently unclear, and could potentially involve differences between cell types (A-431 vs. HCT116). Importantly, using biochemical assays, we demonstrated that the NAA40^S^ isoform has a greater enzymatic efficiency and binding affinity towards its substrates. This observation suggests that the missing N-terminal 21-residue segment of the NAA40^S^ isoform possesses auto-inhibitory activity. It is also interesting to note that NAA40 has a distinct N-terminal domain, which has been previously suggested to assist in the ribosomal binding of the enzyme for co-translational Nt-acetylation [22]. Future studies could also compare the ribosomal binding characteristics of the two isoforms.

## 5. Conclusions

In sum, the human NAA40 gene encodes two enzymatically active isoforms with distinct biochemical and cellular properties, which potentially perform non-redundant functions in normal and cancer settings. We also note the interesting observation of the emergence of a primate-specific transcript encoding specifically NAA40^S^, thus allowing for cells to generate and regulate the abundance of the shorter NAA40 isoform through both translational and transcriptional processes.

## Figures and Tables

**Figure 1 biomolecules-14-01100-f001:**
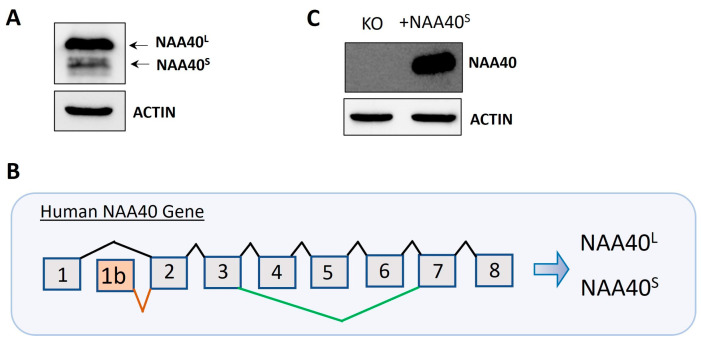
Dual origins of human NAA40^S^. (**A**) Immunoblotting in NAA40^−/−^ HCT116 cells following transfection with plasmid-containing canonical transcripts shows the generation of two distinct NAA40 isoforms at expected sizes. (**B**) Schematic of the human *NAA40* gene and generation of transcripts encoding NAA40^L^ (ENST0000037779) or NAA40^S^ (NM_001300800; ENST00000542163). The *NAA40*^L^ transcript includes eight exons. The NAA40^S^-encoding transcript(s) initiates from an alternative exon 1b, found only in primate species, using a second more downstream translation initiation site to generate a truncated isoform lacking the first 21 amino acids. (**C**) Immunoblotting for NAA40 in HCT116 cells with or without transfection with plasmid-containing NM_001300800 cDNA, which encodes NAA40^S^. Original images can be found in Appendix A.

**Figure 2 biomolecules-14-01100-f002:**
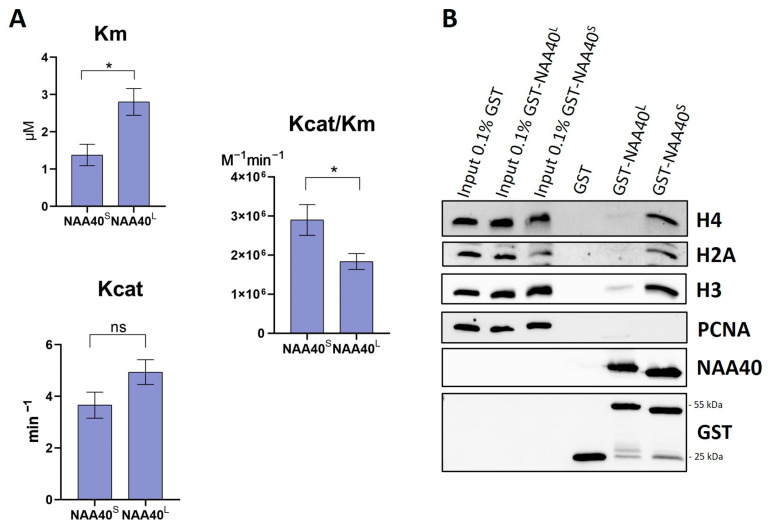
Comparison of enzymatic activities and binding affinities of recombinant NAA40^S^ and NAA40^L^. (**A**) Bar charts displaying K_m_, K_cat_, and K_cat/Km_ as measured in a fluorometric in vitro assay using H4 (1–8) peptide, n = 5, * *p* < 0.05, unpaired *t*-test. (**B**) Comparison of GST Pull down NAA40 isoforms’ binding affinity in nuclear lysates. Weight of proteins: GST tag 25 kDa, GST-NAA40^L^ 55 kDa, GST-NAA40^S^ 25 kDa. Original images can be found in Appendix A.

**Figure 3 biomolecules-14-01100-f003:**
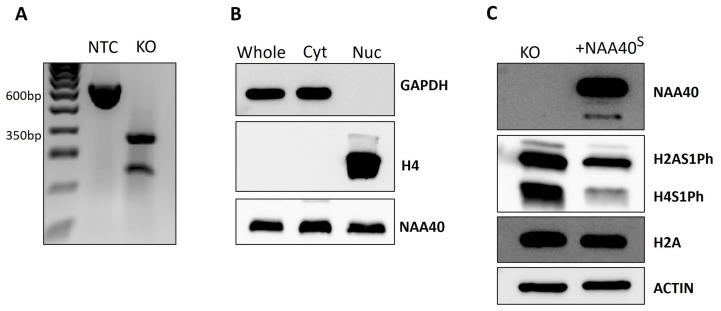
Characterisation of NAA40^S^ in HCT116 cells. (**A**) Agarose gel confirmation of *NAA40* KO. PCR amplification product of size ~600 bp in no template control (NTC) cells corresponds to wild type genomic region that is absent from KO cells. (**B**) Immunoblotting showing the levels of NAA40^S^ in whole, cytosolic, and nuclear cellular compartments in *NAA40* KO cells transfected with NAA40^S^ cDNA. GAPDH and Histone H4 were blotted controls of cytosolic and nuclear compartment purity, respectively. (**C**) Representative immunoblotting showing levels of H2A/H4S1Ph in *NAA40* KO with or without transfection with plasmid-containing NM_001300800 cDNA that encodes NAA40^S^. Original images can be found in Appendix A.

**Figure 4 biomolecules-14-01100-f004:**
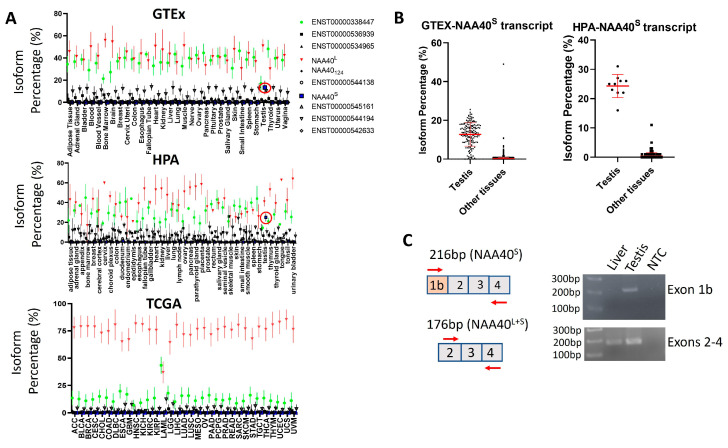
Testis-specific expression of NAA40^S^ coding transcript. (**A**) Plots depicting the median isoform percentage for *NAA40* transcripts in the GTEx, HPA, and TCGA studies. Error bars indicate S.D. The ENST00000377793 (NAA40^L^/NAA40^S^) transcript is depicted as red, non-coding ENST00000338447 as green, and ENST00000542163 (NAA40^S^) as blue. The red circle highlights the ENST00000542163 in the testis. (**B**) Plots comparing the isoform percentage of the transcript encoding NAA40^S^ between the testis and other tissues in GTEX (left) and HPA (right). Error bars indicate S.D., and each dot represents one tissue sample. (**C**) Custom primers were designed for detecting transcripts containing the alternative exon 1b (above) or transcripts containing exons 2–4 (below). PCR was performed on total RNA from a human liver, a human testis, or a non-template control. As expected, PCR amplification of exon 1b was detected only in the testis sample, while exons 2–4 were detected in both tissues.

## Data Availability

Data are contained within the article and Appendix A.

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
