# Peer review of "Biochemical Characterisation of the Short Isoform of Histone N-Terminal Acetyltransferase NAA40"

_biomolecules, 2024, doi:10.3390/biom14091100_

Round 1

Reviewer 1 Report

Comments and Suggestions for Authors

The study reports biochemical characterization of a previously identified shorter variant NAAC40S of N- Alpha Acetyltransferase 40 (NAA40). The study highlights the functional differences between these two NAA40 isoforms through the application of biochemical and cellular assays. They also report that beyond alternative translation, the NAA40S isoform is encoded by a testis-specific transcript.

The research question/hypothesis of the study is clear and the experimental design, methods, and sample size are adequate for drawing significant conclusions. The data has been presented well in text and figures. The findings of the study are biologically significant and supported by relevant literature.  

However, the manuscript will be improved if the following concerns/comments are addressed before publishing the results.

1.      I suggest adding full names of authors on the first page of manuscript instead of using abbreviations.

2.      How many independent NAA40L/S knockout lines were generated? Was the complementation experiment repeated?

3.      RNA from the liver of an individual of 64 years of age and testis of 27 years old, is not good for comparison.

4.      The in vivo enzymatic activity of NAA40 has been carried out with indirect evidence of phosphorylation at serine 1 in histones. Commercial antibodies are available for the acetylated histones, it is advisable to use those antibodies to directly assess the in vivo acetylation activity of the NAC enzymes. The antagonistic effect on serine phosphorylation can be due to other epigenetic marks in the cell as well.  

5.      Lines 114 and 115, reconsider “No target gRNA transfected colonies showed a single band corresponding to ~600 bp and did not show any increase of H2A/H4S1Ph”.

6.      Lines 160 to 162 and lines 175 to 177 create confusion about the isoforms, please clarify.

7.      The manuscript contains non-standard symbols and units in many places. Please ensure that all units and symbols are written consistently in standard format throughout the MS. In addition, there are many typographic mistakes, please thoroughly check the MS for correcting these mistakes. For instance,

a.       line 76, add a space between “100” and “μg/ml” and replace “l” with “L” in “100μg/ml”. Correct the space between values and units and the standard symbol “L” for litre throughout the manuscript.

b.      Line 83, rectify “-20Co” add space between “-20” and “Co” and correct “Co” to “oC”

c.       Line 85, correct “20Mm” to “20 mM”

d.      Line 128, correct “20 Mm Hepes, 1,5 mM MgCl2, 150 mM NaCl, 0,2 mM” to “20 mM Hepes, 1.5 mM MgCl2, 150 mM NaCl, 0.2 mM”

e.       Lines 84, 87, and 122 correct “HCL” to “HCl”

f.        Line 123, “NaCL” to “NaCl”

g.      Line 125, “KCL” to “KCl”

h.      Line 156, delete “isoforms”

i.        Line 255, “bard” to “bars”

j.        Triton X100 is not consistent in lines 101, 123, and 126; hepes in lines 125 and 128

k.      Write uniform NAA40S or NAA40S across the MS

l.        Write all gene names in italic font.

m.    Line 194, add a comma after “purpose” and remove “-” from set-up

n.      Add “and” before “50nM” in line 96 and “0.01%” in line 101

o.      Line 177, remove “s” from “generates”

p.      Line 188, add “it” after “that”

q.      Line 201, rectify “GST-NAA40S 25kDa.”

r.        Figure 2B, replace “,” with “.” in “input 0,1% GST”

s.       Line 274, change “reporter” to “reported”

t.         Line 82, add “was” before “incubated”

8.    Rephrase lines 81—83 to “Cells were lysed by small probe sonication (amplitude 30%, 3 x 45 seconds on ice), and the supernatant was incubated with Glutathione agarose beads (Cytiva 17075601) and washed with ice-cold PBS/1% Triton-X100” or rectify.

9.    Rephrase lines 140—142 to "For the secondary antibody, a horseradish peroxidase (HRP)-conjugated goat anti-rabbit IgG (1:30000, Thermo Scientific) and a polyclonal goat anti-mouse immunoglobulins/HRP (1:1000, Invitrogen) were used" or rectify.

 Regards 

Comments on the Quality of English Language

Though the language and grammar are fine but there are many typographic and format issues, that need a careful revision of the MS for corrections.

Reviewer 2 Report

Comments and Suggestions for Authors

In their manuscript biomolecules-3145428Biochemical characterisation of the short isoform of histone N-2 terminal acetyltransferase NAA40’ the authors Klavaris et al study the functions of alternative forms of N-alpha-acetyltransferase 40 (NAA40), an evolutionarily conserved N-terminal acetyl-13 transferase (NAT) biochemically and in human cells. The studies presented in the manuscript are of high interest for the scientific community.

Major corrections

In the results pg 6 lines 219 to 221 and discussion pg 9 lines 291 to 294 the authors describe the distribution of recombinant NAA40S in cells and direct to Fig. 3C but Fig 3C does not show such cellular distribution in the version submitted for reviewing. Please correct and include the data or delete the comments.

Minor corrections

 A.    Please check the abbreviation used and that they are correctly introduced (if defined in the abstract please repeat in the main text again). Examples NAA40 and NAT are only defined in the abstract.

B.    Paragraph 2.3 line 78, please also include the OD600nm values at which the bacterial cultures were induced.

C.   Please change ‘The reactions were in triplicate …’ to ‘The reactions were carried out in triplicates …’.

D.   Please check the names of suppliers. The company ‘Thermo Fisher Scientific’ is very large and has many subsidiaries but it is introduced as ‘Thermo’, ‘Thermo Fischer’, ‘ThermoScientific’ etc. in the text. Please correct and use the name consistently.

E.    The chemical abbreviation of chloride is ‘Cl’. Please correct.

Comments on the Quality of English Language

The quality of the English language in the manuscript is good. Some very small corrections as included in my review are recommended. 
